# Altered Liver Metabolism, Mitochondrial Function, Oxidative Status, and Inflammatory Response in Intrauterine Growth Restriction Piglets with Different Growth Patterns before Weaning

**DOI:** 10.3390/metabo12111053

**Published:** 2022-11-01

**Authors:** Jun Wang, Pengwei Zhu, Xiaoyu Zheng, Ziwei Ma, Chang Cui, Caichi Wu, Xiangfang Zeng, Wutai Guan, Fang Chen

**Affiliations:** 1Guangdong Province Key Laboratory of Animal Nutrition Control, College of Animal Science, South China Agricultural University, Guangzhou 510642, China; 2State Key Laboratory of Animal Nutrition, College of Animal Science and Technology, China Agricultural University, Beijing 100193, China; 3College of Animal Science and National Engineering Research Center for Breeding Swine Industry, South China Agricultural University, Guangzhou 510642, China

**Keywords:** IUGR, liver, glycolipid metabolism, mitochondria, antioxidation, inflammation

## Abstract

Frequent occurrence of intrauterine growth restriction (IUGR) causes huge economic losses in the pig industry. Accelerated catch-up growth (CUG) in the early stage of life could restore multiple adverse outcomes of IUGR offspring; however, there is little knowledge about this beneficial phenomenon. We previously found that nutrient absorption related to intestinal function was globally promoted in CUG-IUGR piglets before weaning, which might be the dominant reason for CUG, but what this alteration could lead to in subsequent liver metabolism is still unknown. Firstly, a Normal, CUG, and non-catch-up growth (NCUG) piglet model before weaning was established by dividing eighty litters of newborn piglets into normal birth weight (NBW) and IUGR groups according to birth weight, and those piglets with IUGR but above-average weanling body weight were considered CUG, and the piglets with IUGR still below average body weight were considered NCUG at weaning day (d 26). Liver samples were collected and then systematically compared in glycolipid metabolism, mitochondrial function, antioxidant status, and inflammatory status among these three different growth models. Enhanced hepatic uptake of fatty acids, diminished de novo synthesis of fatty acids, and increased oxidation of fatty acids were observed in CUG livers compared to Normal and NCUG. In contrast, the NCUG liver showed enhanced glucose uptake and gluconeogenesis compared to Normal and CUG. We also observed deteriorating hepatic vacuolation in NCUG piglets, while increasing hepatic lipid deposition in CUG piglets. Besides, the expression of genes related to mitochondrial energy metabolism and biogenesis was reduced in CUG piglets and the phosphorylation level of AMPK was significantly higher compared to Normal (*p* < 0.05). Moreover, NCUG liver showed decreased T-AOC (*p* < 0.01) and GSH-PX (*p* < 0.05), increased MDA concentrations (*p* < 0.01), upregulated phosphorylation levels of ERK and NF-κB (*p* < 0.05), and elevated pro-inflammatory factors IL-1β, IL-6 and TNF-α (*p* < 0.05) compared to Normal. Furthermore, correlation analysis revealed a significant positive correlation between glucose metabolism and inflammatory factors, while a negative correlation between mitochondrial function-related genes and fatty acid transport. NGUG piglets showed simultaneous enhancement of glucose uptake and gluconeogenesis, as well as reduced antioxidant capacity and increased inflammatory status, whereas CUG comes at the expense of impaired hepatic mitochondrial function and pathological fat accumulation.

## 1. Introduction

Intrauterine growth retardation (IUGR), defined as impaired growth and development of the mammalian embryo/fetus or its organs during pregnancy, is a major inducer of high pre-weaning morbidity and mortality in the pig industry [1]. IUGR is usually accompanied by impaired organ development and metabolic disorders which can result in long-term adverse impacts on their postnatal life [2]. It was observed that some IUGR offspring could undergo a period of accelerated growth and eventually reach a normal weight, which is called catch-up growth (CUG) [3,4]. Accumulating evidence shows that CUG eliminates multiple growth defects of IUGR, including growth retardation, and thus reduces the incidence of malnutrition, mortality, and morbidity [5]. Although the benefit of CUG may be of great significance in ameliorating the economic losses caused by the high occurrence of IUGR in the pig industry, there have only been very limited investigations paying attention to this subject.

During pregnancy, IUGR fetuses preferentially supply nutrients to vital organs such as the brain and heart to firstly ensure basic survival, which would directly damage other peripheral organs due to insufficient nutrient supply in utero [6,7]. As the largest metabolic organ, the liver plays an important role in macronutrient metabolism, immune regulation, and the maintenance of lipid and cholesterol homeostasis [8,9]. It has been widely reported that IUGR piglets are usually accompanied by liver dysfunction, which is characterized by morphological damage, metabolic disorders, and changes in abnormal blood lipid levels [10,11]. Investigations in humans, mice, and rats have shown that IUGR offspring might exhibit CUG when the postnatal nutritional environment is improved, while this change in nutritional status potentially caused a severe metabolic burden on the liver resulting in abnormal glucose metabolism, lipid deposition, and lipid degeneration [12,13], as well as mitochondrial dysfunction [14,15]. As a consequence, CUG-IUGR in adulthood has been shown to have a high incidence rate of diseases related to functional liver disorders, such as fatty liver [16], obesity, insulin resistance [10], poor glucose tolerance, and so on [16]. Our previous study revealed that nutrient absorption related to intestinal function was globally promoted in CUG-IUGR piglets before weaning which might be the dominant reason for CUG [17], but what this alteration could lead to in subsequent liver metabolism was still unknown.

Therefore, in the present study, we systematically compared the liver glycolipid metabolism, mitochondrial function, antioxidant capacity, and inflammatory status of CUG, NCUG, and Normal before weaning to study the effects of different growth patterns on liver metabolism and function in IUGR piglets. To our best knowledge, this is the first study to use weanling piglets as CUG models, and the results would better fill the understanding gap regarding the consequences and underlying mechanisms of pig early-stage CUG.

## 2. Materials and Methods

### 2.1. Animals, Experimental Diets, and Sample Collection

During gestation, 80 healthy pregnant sows (Landrace Yorkshire) were preselected according to similar parity (2–3), litter size, and expected farrowing date. The newborn piglets were weighed immediately after delivery. IUGR piglets were defined as having an average birth weight two standard deviations below the average birth weight of the total population, and normal birth weight piglets were defined as having a birth weight within 0.5 standard deviations of the average birth weight of the total population. Based on this criterion, 28 normal-weight piglets and 55 IUGR piglets were selected, and all the piglets were weighed and measured weekly. All piglets were fed milk and water freely. The sows were fed the same composition and nutrient content basal diet.

The composition of the basal diet (Table 1) was formulated in compliance with the National Research Council (1998) nutrient requirements. Sows were housed in individual stalls (2.0 m × 0.6 m) and fed twice (07:30 and 15:30) a day with a constant amount of 2.5 kg during gestation. Five days before farrowing, the sows were moved into individual farrowing crates (2 m × 1.5 m), with parturitions being observed frequently in all groups. The farrowing room was strictly controlled, with disturbances avoided as much as possible, and the inner temperature was kept approximately at 20 to 25 °C by an air conditioning system. All the farrowing crates were equipped with a feeder and a nipple drinker for sows; a nipple drinker and a heating lamp were provided for suckling piglets. Sows were not provided with feed on their farrowing day but were fed a lactation diet twice a day (07:30 and 17:00) from the next day until weaning. The initial amount of feed was 1.5 kg on the first day postpartum which increased daily by 0.5 kg until 7 d postpartum, then the sows were fed ad libitum until weaning. On weaning day (26 days), 6 IUGR piglets exceeding the weaning average individual weight were divided into the CUG group, and 6 IUGR not exceeding the weaning average individual weight were divided into the NCUG group (Figure 1). Meanwhile, 6 weaning piglets with normal birth weight were allotted to Normal group (Figure 1). Animals were euthanized by humane euthanasia with sodium pentobarbital. The weights of the heart, liver, spleen, lung, and kidney from piglets of each group were recorded. Liver specimens were collected from the same position in the left lobe of the liver and immersed in a 4% paraformaldehyde solution for preservation. In addition, liver tissue samples were collected, snap-frozen in liquid nitrogen, and stored in a −80 °C freezer.

### 2.2. Liver Biopsy Staining

Liver specimen samples were processed using standard paraffin-embedding techniques. After dehydration, liver samples were sectioned at 5 μm and stained with hematoxylin and eosin (H&E) (Thermo Fisher Scientific, Beijing, China) for histopathological evaluation. Stained sections were observed and photographed under a light microscope (Nikon YS100; Nikon Corporation, Tokyo, Japan).

### 2.3. Oil Red O Staining of Liver Sections

Liver tissue specimens were removed from the fixative, placed in a 15% sucrose solution, dehydrated, and settled in a 4 °C refrigerator, and then transferred to a 30% sucrose solution to dehydrate and settle in a 4 °C refrigerator. The dehydrated tissues were blotted with filter paper to dry the surface water and then dripped with OCT embedding agent (Sakura, Tokyo, Japan), sectioned (5 μm thickness) while using a frozen microtome (Thermo Fisher Scientific, Beijing, China), and then stained using oil red stain for 8 to 10 min, followed by background differentiation using isopropyl alcohol and then hematoxylin (Thermo Fisher Scientific, Beijing, China) staining. The prepared sections were observed and photographed using a light microscope (Nikon YS100; Nikon Corporation, Tokyo, Japan) equipped with a digital camera (NIKON DS-Fi1c, Tokyo, Japan).

### 2.4. Hepatic Oxidative Stress

To evaluate hepatic oxidative stress, the amounts of malondialdehyde (MDA, A003-1), reduced glutathione (GSH, A006-2-1), and the activities of total superoxide dismutase (T-SOD, A001-3), total antioxidant capacity (T-AOC, A015-3-1), and glutathione peroxidase (GSH-PX, A005-1) were measured using commercially available kits from Nanjing Jiancheng Bioengineering Institute (Nanjing, China), according to the guidelines of the manufacturer.

### 2.5. Total RNA Isolation and Real-Time PCR

Total RNA was isolated from 100 mg tissue samples using TRIZOL reagent (Thermo Fisher Scientific, Beijing, China) according to the manufacturer’s instructions. The RNA quality was confirmed by calculating the OD 260/280 ratio via an RNA Concentration Analyzer (NanoDrop, 2000c; Thermo Scientific, Wilmington, NC, USA), and the integrity of the RNA was verified by agarose gel electrophoresis. RNA was reverse transcribed to cDNA using gDNA Remover and 4× RT Master Mix (EZBioscience, Beijing, China). 2× Color SYBR Green qPCR Master Mix (EZBioscience, Beijing, China) was used for quantitative PCR. PCR reactions were performed with an Mx3005p instrument (Stratagene, San Diego, CA, USA). Primer sequences for qRT-PCR are listed in Table 2. Gene expression profiles were normalized with gene expression profiles of β-actin, and the relative mRNA expression of target genes was calculated using the 2-ΔΔCt method.

### 2.6. Western Blot Analysis

Liver tissue samples were homogenized in RIPA lysis buffer (Beyotime, Nanjing, China). Extracted protein samples were heated in water at 100 °C for 5 min and separated by electrophoresis (Bio-Rad, Richmond, CA, USA) in 8-12% SDS-PAGE gel, and then electrically transferred to polyvinylidene difluoride (PVDF) membranes (Millipore, Billerica, MA). After polyvinylidene fluoride membranes were blotted with Tris-buffer containing 5% fat-free dry milk and 0.05% Tween-20 (Sangon biotech, Shanghai, China) for 1 h at 25 °C, they were rinsed in TBST (TBST; 0.05% Tween 20, 100 mmol/L of Tris-HCl, and 150 mmol/L of NaCl, pH 7.5) (Sangon biotech, Shanghai, China) 4 times and incubated overnight at 4 °C with primary antibodies: GLUT1 (Abcam, ab150299, Cambridge, UK) dilution ratio of 1:500. The dilution ratios of p-AMPK (Cell Signaling Technology, 2535T, Boston, USA), AMPK (Abcam, ab3759, Cambridge, UK), NF-κB (Proteintech, 20536-1-AP, Chicago, IL, USA), p-NF-κB (Cell Signaling Technology, 3033s, Boston, USA) were all 1:1000. The dilution ratios of ACACA (Abcam, ab72046, Cambridge, UK), FASN (Abcam, ab99359, Cambridge, UK), and β-actin (Proteintech, 20536-1-AP, Chicago, FL, USA) were all 1:2000. The dilution ratios of JNK (Cell Signaling Technology, 9252s, Boston, USA), p-JNK (Cell Signaling Technology, 4668s, Boston, USA), ERK (Cell Signaling Technology, 9102s, Boston, USA), p-ERK (Cell Signaling Technology, 9101s, Boston, USA) were all 1:3000. After washing, membranes were incubated with a secondary antibody (Amersham Biosciences, Buckinghamshire, UK). Protein expression was measured and analyzed using a Fluor Chem M fluorescent imaging system (Protein Simple, Santa Clara, CA, USA) and ImageJ software.

### 2.7. Statistical Analysis

Data were collected from at least three independent experiments, and all data were analyzed using one-way analysis of variance (ANOVA) followed by Duncan’s multiple range test in SPSS software (v20.0; SPPS Inc., Chicago, IL, USA). Data are shown as means ± SEM. *p* < 0.05 was considered statistically significant, and *p* < 0.01 was considered highly statistically significant. Liver antioxidant capacity index parameters measured by kits and liver glucose metabolism, inflammatory status, mitochondrial function, and fatty acid metabolism-related gene mRNA expression level index parameters measured by RT-PCR were tested using Pearson’s correlation test (all groups), and a correlation heat map was drawn using the heat map R package.

### 2.8. Ethics Approval and Consent to Participate

All animal procedures used in this experiment were approved by the Animal Care and Use Committee of the South China Agricultural University (No. 20110107-1, Guangzhou, China).

## 3. Results

### 3.1. Growth Performance and Organ Index of Piglets with Different Growth Patterns

To establish animal models with different growth patterns, body weight and length were measured weekly to track the growth pattern of piglets. Body weight, length, and BMI of CUG and NCUG were significantly lower than Normal at birth (*p* < 0.05), while all these parameters of CUG have no obvious difference from the Normal group on weaning day. Additionally, the body weight, body length, and BMI of NCUG piglets at 1, 14, 21, and 26 days were significantly lower than those of Normal piglets, showing growth retardation (*p* < 0.05) (Figure 2). Mortality of IUGR piglets increased most rapidly in the first week and reached its highest rate in the second week (14.55%). At the end of the experiment, piglets achieving CUG accounted for 25.45% of IUGR piglets (Figure 2D). The average daily gain (ADG) during the whole suckling period had a higher trend in CUG compared with the Normal group (Table 3), and the ADG in the third week of suckling showed significantly higher in CUG than Normal (*p* < 0.01) (Table 4). The weight of the spleen (*p* < 0.05), heart (*p* < 0.01), kidney (*p* < 0.01), lung (*p* < 0.01), and liver (*p* < 0.01) were significantly lower in NCUG than in Normal, while the liver weight was significantly lower in CUG than in Normal (*p* < 0.05), other organs were not significantly different from Normal (Table 5). The organ index was obtained by dividing the organ weight by the body weight, liver organ index decreased in CUG compared to NCUG (*p* < 0.05), while kidney organ index was greatest in NCUG compared to CUG and Normal (*p* < 0.05), and all other organs were proportionally reduced with body weight (Table 6).

### 3.2. Liver Histology Section

We subsequently detected hepatic pathological sections in three piglet groups. Compared to Normal and CUG piglets, NCUG piglets displayed apparent vacuoles, and severe structural damage appeared in the hepatocytes (Figure 3). These results confirmed the existence of a liver injury in NCUG piglets.

### 3.3. Hepatic Lipid Metabolism in Piglets with Different Growth Patterns

Oil Red O staining was performed to estimate lipid content in hepatocytes. CUG piglets showed significant lipid deposition in the liver, but no obvious lipid droplets were observed in either Normal or NCUG groups (Figure 4A). In addition, fatty acid transporters and genes involved in lipid metabolism were assayed to determine the source of CUG hepatic lipid deposition in the present study. The results showed that the mRNA expression of *FATP2* and *CD36* in CUG were significantly higher than those in Normal (*p* < 0.05), and *FATP5* was significantly higher than that in the NCUG group (*p* < 0.05). There was no significant difference observed between NCUG and Normal piglets (Figure 4B). Interestingly, significant downregulation of fatty acid de novo synthesis-related proteins was also observed in CUG. Compared with Normal, a significant decrease in the protein levels of ACACA and FASN was observed in CUG (*p* < 0.05), while only a decrease in ACACA was observed in NCUG (*p* < 0.05) (Figure 4D). Expression of *PPAR-γ* (*p* < 0.01), *SREBP1* (*p* < 0.05), *ACACA* (*p* < 0.01), *FASN* (*p* < 0.01), and *LPIN1* (*p* < 0.05) decreased in CUG livers compared with Normal and NCUG. In addition, the expression of fatty acid de novo synthesis-related genes in NCUG was not significantly different from Normal (Figure 4C,E). In addition, the mRNA expression level of *ACL* was extremely significantly higher in NCUG than in Normal and CUG (*p* < 0.001), and there was no significant difference between CUG and Normal (Figure 4E). Expression of the fatty acid oxidation regulatory gene *PPAR-α* was upregulated in CUG compared with NCUG and Normal (*p* < 0.05), while no significant difference between NCUG and Normal was observed (Figure 4C). The expression of *CPT1* in CUG was significantly upregulated compared with NCUG (*p* < 0.05), but was not significantly different from Normal (Figure 4E).

### 3.4. Hepatic Glucose Metabolism in Piglets with the Different Growth Patterns

Compared with CUG and Normal, the protein level of GLUT1 was significantly increased in NCUG (*p* < 0.05) (Figure 5A), along with the mRNA expression levels of *GLUT1* and GLUT2 (*p* < 0.01) (Figure 5B,C). In CUG, no significant difference was observed in GLUT1 protein level and mRNA level compared with Normal (Figure 5B), while the mRNA level of *GLUT2* was significantly lower than that in NCUG (*p* < 0.001) and Normal (*p* < 0.01) (Figure 5C). The expression of *GLUT4* in CUG was significantly lower than that in NCUG and Normal (*p* < 0.01), and there was no significant difference between NCUG and Normal (Figure 5D). Compared with Normal, the mRNA expressions of *G6PC*, *PEPCK*, and *GCK* genes were significantly increased in CUG (*p* < 0.05) and extremely significantly increased in NCUG (*p* < 0.01). Although there was no significant difference between NCUG and CUG in *G6PC* and *GCK*, *PEPCK* of NCUG was significantly upregulated compared to CUG (*p* < 0.05). Compared to Normal, *PFKL* was significantly upregulated in NCUG (*p* < 0.05), and there was no significant difference between CUG and Normal (Figure 5E).

### 3.5. Mitochondrial Electronic Transmission Chain (ETC) and AMPK Signaling Pathway in the Liver among Different Growth Patterns of Piglets

We measured the expression of genes related to ETC and biogenesis to compare the effect of CUG growth on mitochondrial function. Gene expression of *NDUFA1* and *NDUFA6* decreased in CUG compared with NCUG (*p* < 0.01), but was not significantly different from Normal. The expression of *NDUFA13* decreased in CUG compared with NCUG and Normal (*p* < 0.05), while the expression of *NDUFB1* in CUG was significantly lower than that in Normal (*p* < 0.05) and extremely significantly lower than that in NCUG (*p* < 0.01). Compared to NCUG and Normal, CUG reduced the gene expression of *UQCRB* (*p* < 0.01) and *Cytc* (*p* < 0.05). In addition, the expression of *COX IV* and *COX V* was downregulated in the CUG compared to Normal (*p* < 0.05) (Figure 6A,B). Furthermore, CUG inhibited the expression of gene *TFAM* related to mitochondrial biogenesis (*p* < 0.05) (Figure 6C). In contrast, no significant difference in mitochondrial energy metabolism and biogenesis-related genes was observed between NCUG and Normal. In addition, compared with Normal, CUG piglets increased the ratio of phosphorylated AMPK to total AMPK (p-AMPK/AMPK) (*p* < 0.05), while p-AMPK/AMPK in NCUG was not significantly different from that of CUG and Normal (Figure 6D).

### 3.6. Hepatic Oxidative Stress Comparison among Different Growth Patterns of Piglets

Given the significant differences in liver metabolic indices between CUG and NCUG, we next measured the differences in antioxidant capacity between the three groups of piglets. In NCUG, the activities of T-AOC (Figure 7A) and GSH-PX (Figure 7C) were significantly decreased compared with CUG and Normal (*p* < 0.05), while the MDA content was significantly increased compared with CUG and Normal (*p* < 0.01) (Figure 7B). No significant difference in GSH (Figure 7D) and T-SOD (Figure 7E) was observed between NCUG and CUG groups. The mRNA expression of *Nrf2* was significantly increased in CUG (*p* < 0.05) and extremely significantly increased in NCUG (*P* < 0.01). Compared with Normal, the mRNA expression levels of *SOD1* and *GPX4* were significantly upregulated in NCUG (*p* < 0.05), while no significant difference was observed between CUG and Normal (Figure 7F).

### 3.7. Hepatic Inflammatory State and NF-κB/MAPK Signaling Pathway

There are no significant differences in the phosphorylation of JNK protein among Normal, CUG and NCUG (Figure 8A). The phosphorylation of ERK protein in NCUG and CUG was significantly higher than that in Normal (*p* < 0.01) (Figure 8A). The level of NF-κB phosphorylation was significantly increased in NCUG compared with Normal (*p* < 0.05), while no significant difference was observed between CUG and Normal (Figure 8A). The mRNA expressions of *NF-κB* and MAPK-related proteins and their downstream pro-inflammatory factors were consistent with the phosphorylation of *NF-κB*/MAPK pathway proteins. *NF-κB* was significantly upregulated in NCUG compared to Normal (Figure 8B). The mRNA expressions of *IL-1β* and *IL-6* in NCUG were significantly higher than those in CUG and Normal (*p* < 0.01), and the expression of *TNF-α* was significantly higher than in Normal (*p* < 0.05). In contrast, no significant difference was observed between the expression of pro-inflammatory factors in Normal and CUG (Figure 8C).

### 3.8. Correlation Analysis of Liver Metabolism-Related Indexes

Fatty acid transport-related genes (*FATP2*/*FATP4*/*FATP5*/*CD36*) and fatty acid oxidation-related genes (*PPAR-α*/ *CPT1*) were significantly negatively correlated with fatty acid de novo synthesis-related genes (*PPAR-γ*/*SREBP1*/*ACACA*/*FASN*/*LPIN*) (R > 0.5, *p* < 0.05) (Figure 9C). However, hepatic glucose transport (*GLUT1*/*GLUT2*), gluconeogenesis and glycolysis-related genes (*G6PC*/*PEPCK*/*GCK*/*PFKL*/*PKL*) were associated with pro-inflammatory factor (*IL-1β*/*IL-6*/*IL-12*/*TNF-α*) expression with a significantly positively correlation (R > 0.5, *p* < 0.05) (Figure 9B). In addition, the gene expression levels of *PGC-1α-*, *TFAM-*, and ETC-related proteins were significantly negatively correlated with those genes related to fatty acid transport and fatty acid oxidation, while genes related to fatty acid de novo synthesis and glucose transport showed a significantly positive correlation (Figure 9D).

## 4. Discussion

Postnatal catch-up weight gain in IUGR individuals is primarily due to greater lipid accumulation in organs and tissues rather than muscle gain [18]. It has been demonstrated that IUGR with a CUG pattern is highly associated with an increased risk of nonalcoholic fatty liver disease (NAFLD), obesity, and metabolic syndrome in adulthood [10,19]. In the current study, we also observed a significant accumulation of lipid droplets in the liver of the CUG group, which is consistent with previous results. Generally, hepatic lipid deposition is the result of fatty acid influx over efflux under the condition of disordered lipid metabolism [20]. The results of our current study showed that the expression of several key genes related to fatty acids transporters including fatty acid transport protein (*FATP2*, *FATP5*) and fatty acid translocase (*CD36*) was higher in the liver of the CUG group than in those of the Normal and NCUG groups, which indicated more fatty acid uptake from circulation for potential triglyceride (TG) synthesis or oxidation. However, we interestingly found that the genes related to TG de novo syntheses such as peroxisome proliferator-activated receptor gamma (*PPAR-γ*), acetyl-CoA carboxylase alpha (*ACACA*) and fatty acid synthase (*FASN*) decreased in CUG compared with NCUG and Normal group. Evidence from rodent stable isotope analysis studies showed that the amount of esterified fatty acids in the liver is 2 times greater than the de novo synthesis of fatty acids [21]. Similar results were found in a human study that circulating uptake accounted for 45.1% to 74.3% of hepatic triacylglycerol (TAG) sources, while only 12.7% to 37.0% of fatty acids were de novo synthesized, suggesting that exogenous lipid uptake is the main source of hepatic lipid deposition [22]. Therefore, we propose that the increased lipid deposition in the liver of CUG piglets is mainly due to increased hepatic fatty acid uptake rather than increased lipid de novo synthesis. It has been well demonstrated that peroxisome proliferator-activated receptor alpha (*PPAR-α*) could be activated by elevated lipid accumulation in the liver, and then in turn induce the expression of genes involved in fatty acids oxidative pathway to eliminate excess hepatic lipid store [23]. Compensatory enhancement of fatty acid oxidation was observed in the early stage of NALFD formation to maintain hepatic lipid homeostasis in a previous investigation [20]. Consistently, we also found in the present study that the fatty acid oxidation-regulated nuclear receptor *PPAR-α* and fatty acid β oxidation key enzyme (*CPT1*) were upregulated in CUG compared with NCUG, implying that adaptive increased fatty acid oxidation was not sufficient to reverse hepatic lipid deposition.

Data from several studies suggest that IUGR fetuses have lower glucose concentration levels than Normal piglets [24,25]. It is worth noting that the liver, as an insulin-responsive organ, exhibited an adaptive promoted glucose storage capacity with elevated insulin action, glucose production, and uptake, while reduced glycogen synthesis to maintain glucose homeostasis under the hypoglycemic–hypoinsulinemic IUGR environment during pregnancy [24]. Interestingly, many recent studies have shown that when CUG occurs, IUGR offspring exhibit decreased glucose tolerance [26]. In the present experiment, hepatic glucose uptake key proteins (GLUT1, GLUT2) were upregulated in NCUG compared to Normal, and relatively, hepatic glucose transport proteins (*GLUT1*, *GLUT2*, *GLUT4*) expression was downregulated in CUG compared to NCUG, which may be one of the reasons for the reduced glucose tolerance in CUG-IUGR offspring. In addition, several studies have shown that IUGR has early activation of hepatic gluconeogenesis to adapt to the intrauterine hypoglycemic environment [27] and resistance to insulin’s normal suppression of hepatic gluconeogenesis [28]. Similar results have been observed in a pig model, where IUGR consistently increases the pig’s hepatic gluconeogenic key enzymes (*PEPCK* and *G6PC*) expression and activities at 49 and 105 days [29,30]. Another study found that dietary restriction further increased the expression of hepatic *PEPCK* in adult IUGR pigs, and suggested that these alterations may exacerbate glucose intolerance in adult offspring exposed to intrauterine undernutrition [31]. Consistent with these results, we found that *PEPCK* and *G6PC* expression were enhanced in IUGR piglets, and *PEPCK* expression was further upregulated in NCUG compared to CUG, suggesting that NCUG has stronger gluconeogenesis. Several studies have shown that CUG-IUGR offspring manifested elevated fasting blood glucose levels [32]. In addition, combined with the results of our other experiments, we found elevated glucose transporter in the intestine of CUG, which may indicate that early nutritional correction attenuates abnormal gluconeogenesis in CUG (unpublished results). In summary, glucose uptake, glycolysis, and gluconeogenesis were more significantly enhanced in NCUG. Brown et al. reported that this may create a potential futile cycle in glucose metabolism in the IUGR offspring liver [33], which may be associated with retarded growth of NCUG.

Mitochondria acting as a metabolic hub and energy-producing factory are involved in a variety of vital cellular activities [34]. Mitochondrial energy production is mainly accomplished by the electron transport chain (ETC) complexes (CI, CII, CII, CIV, CV) located on the mitochondrial inner membrane [34]. Evidence is mounting that poor intrauterine nutrition causes long-lasting adverse effects in hepatic mitochondria biogenesis and function that may contribute to postnatal development [12]. IUGR has been reported to be associated with compromised mitochondrial function showing less ATP content, reduced complex activity, and downregulated expression of genes related to mitochondrial biogenesis and energy metabolism [35]. Previous research interestingly reported that CUG-IUGR rats induced by nutritional treatment lead to further deterioration of mitochondrial dysfunction with higher oxidative injury and decreased complex II and mitochondrial transcription factor A (*TFAM*) [36]. In another similar study, decreased peroxisome proliferator-activated receptor-γ coactivator 1α (*PGC-1α*) mRNA expression and mtDNA/nDNA ratio were observed in the liver of CUG-IUGR rats, suggesting reduced mitochondrial content and impaired mitochondrial function [37]. In the current study, we consistently found that the mRNA levels of ETC complex subunits and *TFAM* were significantly reduced in CUG pigs, while there was no difference between NCUG pigs and Normal pigs. Serine/threonine kinase adenosine monophosphate-activated protein kinase (AMPK) is a low ATP sensor that restores intracellular ATP homeostasis [38]. In addition, an increasing piece of evidence points to inhibition of the mitochondrial ETC complex to induce AMPK activation [39]. We also observed that the activation of the AMPK signaling pathway was significantly upregulated in the CUG-IUGR piglet’s liver compared with the other two groups in the current study, further confirming the mitochondria impairment.

IUGR offspring are characterized by systemic redox imbalance and impaired antioxidant capacity, leading to apparent oxidative stress and tissue damage [40,41], especially in the liver due to its extensive energy metabolism [41,42]. Several researchers have observed decreased T-AOC and SOD activity and increased MDA (a marker of lipid peroxidation) [43] concentration in the liver of IUGR piglets compared to normal piglets [41]. Interestingly, we found that NCUG-IUGR had elevated concentrations of MDA and decreased T-AOC and GPX enzyme activities compared with Normal and CUG piglets indicating higher oxidative stress and reduced antioxidant capacity in the present study. Meanwhile, no significant difference was observed between the Normal and CUG-IUGR groups, implying a restored redox balance in the CUG liver. The Nrf2 system has been described as a critical defense mechanism for animals against oxidative stress [44]. Under oxidative stress with cellular excess ROS accumulation, the interaction between Nrf2 and Keap1 is disrupted and then release-activated *Nrf2* translocation into the nucleus to bind with ARE sequence, consequently resulting in numbers of antioxidative gene expression to counteract the adverse redox imbalance [45,46]. In the present investigation, we also found that the transcript levels of *Nrf2* and its downstream target genes including *SOD1* and *GPX4* were greatly elevated in NCUG piglet livers which were consistent with their higher oxidative stress.

Additionally, there is considerable evidence demonstrating that oxidative stress is involved in chronic inflammation [47] by activating survival signaling pathways such as the Nuclear factor kappa-B (NF-κB) [48,49] and mitogen-activated protein kinase (MAPK) [50] signaling pathways, which in turn promote the transcription of *TNF-α*, *IL-1β*, *IL-6* and *IL-12* thereby promoting the development of inflammation [51]. A growing number of studies show that IUGR leads to a high risk of fetal liver inflammation [52,53]. Research has demonstrated that the concentration and expression of pro-inflammatory cytokines *IL-1β*, *IL-6*, and *TNF-α* in the liver and serum were enhanced in IUGR piglets [35,54]. The results of the present study showed that gene expression of *TNF-α*, *IL-1β*, and *IL-6* in the liver of NCUG piglets was significantly higher than in Normal piglets, and the NCUG liver exhibited a more pronounced vacuolated structure, which might be attributed to the extensive oxidative stress and suggested that NCUG-IUGR piglets were prone to liver inflammatory injury. It is worth noting that the upregulated activation of the ERK signaling pathway was observed in the livers of both NCUG and CUG piglets, which was different from the result that oxidative indicators were only changed in NCUG, but the NF-κB signal showed further activated in NCUG groups compared to Normal. These results suggest that NF-κB pathways might play the dominant role in chronic inflammatory injury induced by oxidative stress in IUGR piglets, implying a potential intervention target for liver dysfunction caused by IUGR in the future. In addition, we found that abnormal liver glucose metabolism in IUGR piglets was significantly and positively correlated with pro-inflammatory factors in the correlation analysis of this experiment, suggesting that abnormal glucose metabolism in NCUG may be associated with an elevated inflammatory state in the liver (Figure 9B).

Although we found comprehensive alterations in CUG piglets’ livers, including lipid and glucose metabolism, oxidative stress, and inflammation response, as well as related regulatory signaling pathway from the present study, we still have no idea how these differences happen and what the underlying mechanisms for these differences are, which is of importance for understanding the occurrence of CUG before weaning. Additionally, it is worth further exploring liver alteration after weaning to clarify the long-term impact of different growth patterns on liver metabolism and function in the future.

## 5. Conclusions

The present study is the first to establish naturally occurring CUG and NCUG-IUGR piglet models before weaning to investigate the hepatic alteration with different growth patterns. Obviously, hepatic metabolism, mitochondrial function, oxidative status, and the inflammatory response showed comprehensive differences. The liver of NCUG-IUGR piglets exhibited elevated glucose uptake and gluconeogenesis, increased expression of pro-inflammatory factors, and lower antioxidant capacity. In contrast, hepatic mitochondrial function was impaired and fatty acid uptake was enhanced resulting in excess fat accumulation in CUG-IUGR piglets. These results shed light on and deepen our understanding of the occurrence of CUG in IUGR piglets before weaning. Therefore, targeted nutritional intervention strategies should consider the trade-off between postpartum catch-up growth and negative metabolic effects in IUGR piglets to provide a theoretical basis for alleviating the high morbidity and mortality and permanent retardation of growth in IUGR piglets prior to weanling.

## Figures and Tables

**Figure 1 metabolites-12-01053-f001:**
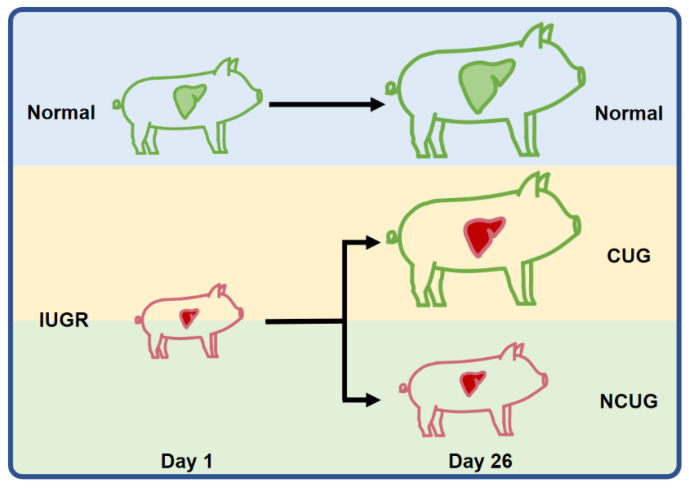
Overview of the experimental design. Based on their birth weight, the piglets were classified into either the IUGR or Normal group. In addition, based on weaning weight, IUGR piglets were divided into the CUG group and NCUG group. CUG = catch-up growth; NCUG = non-catch-up growth; WW = weaning weight.

**Figure 2 metabolites-12-01053-f002:**
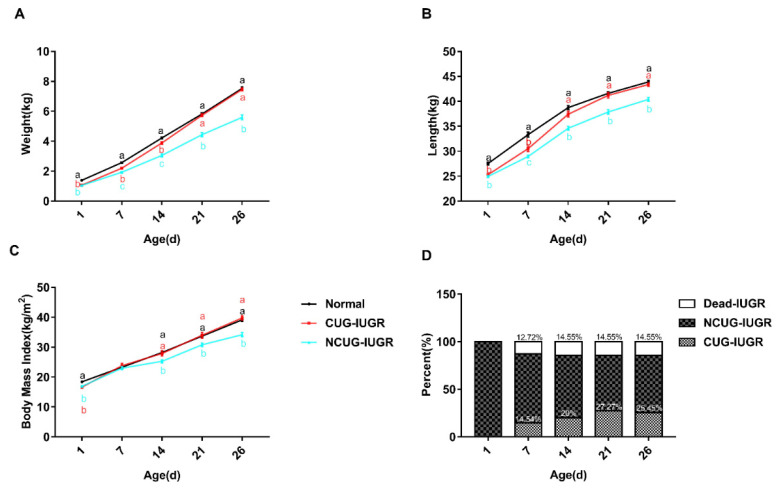
Growth performance and organ index of piglets. (**A**) Body weight, (**B**) body length, (**C**) body mass index, (**D**) mortality rate, and percentage of CUG before weanling. a, b, c Different letters indicated a significant difference (*p* < 0.05). CUG = catch-up growth; NCUG = non-catch-up growth; WW = weaning weight.

**Figure 3 metabolites-12-01053-f003:**
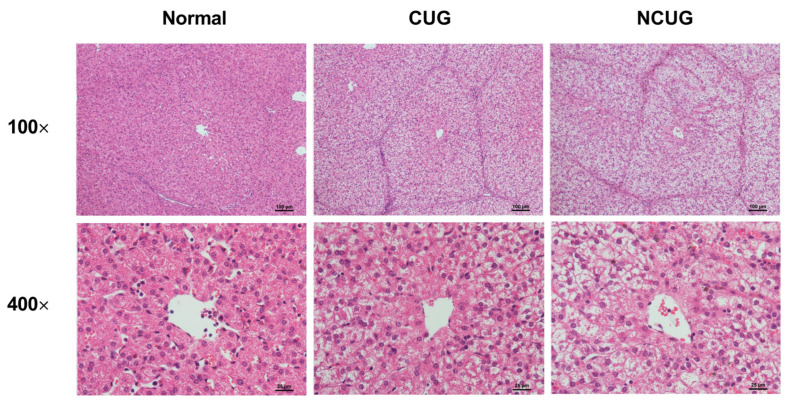
Histopathology with H&E staining of the liver section in different groups (Normal, CUG, NCUG). CUG = catch-up growth; NCUG = non-catch-up growth; WW = weaning weight.

**Figure 4 metabolites-12-01053-f004:**
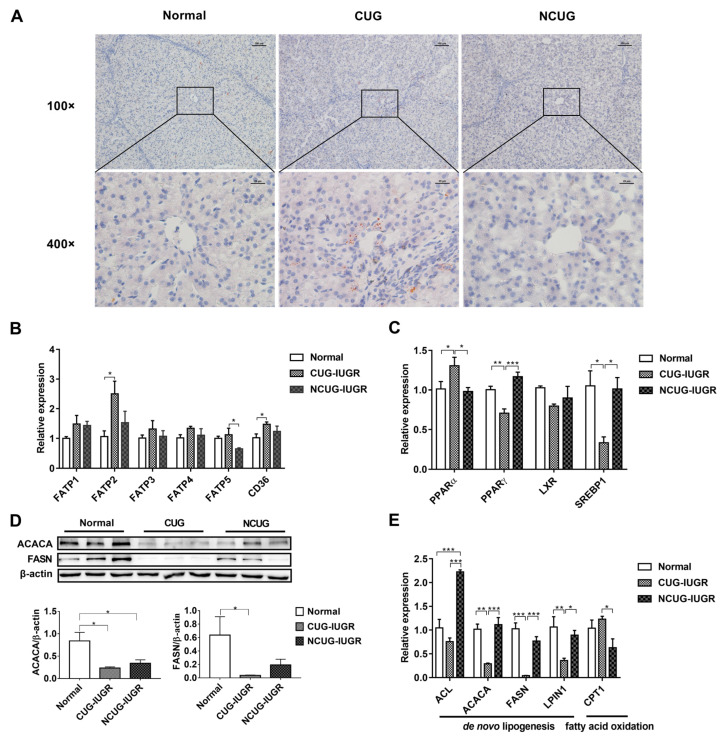
Effects of CUG and NCUG growth patterns on liver lipid metabolism in piglets. (**A**) Piglet liver oil red O staining. (**B**) mRNA expression of liver fatty acid transport-related genes. (**C**) mRNA expression of nuclear receptor genes regulates hepatic lipid metabolism. (**D**) Protein levels of lipid synthesis-related proteins ACACA and FASN. (**E**) mRNA expression of genes related to hepatic lipid metabolism. All data with error bars are mean ± SEM. * *p* < 0.05, ** *p* < 0.01, *** *p* < 0.001. CUG = catch-up growth; NCUG = non-catch-up growth; WW = weaning weight.

**Figure 5 metabolites-12-01053-f005:**
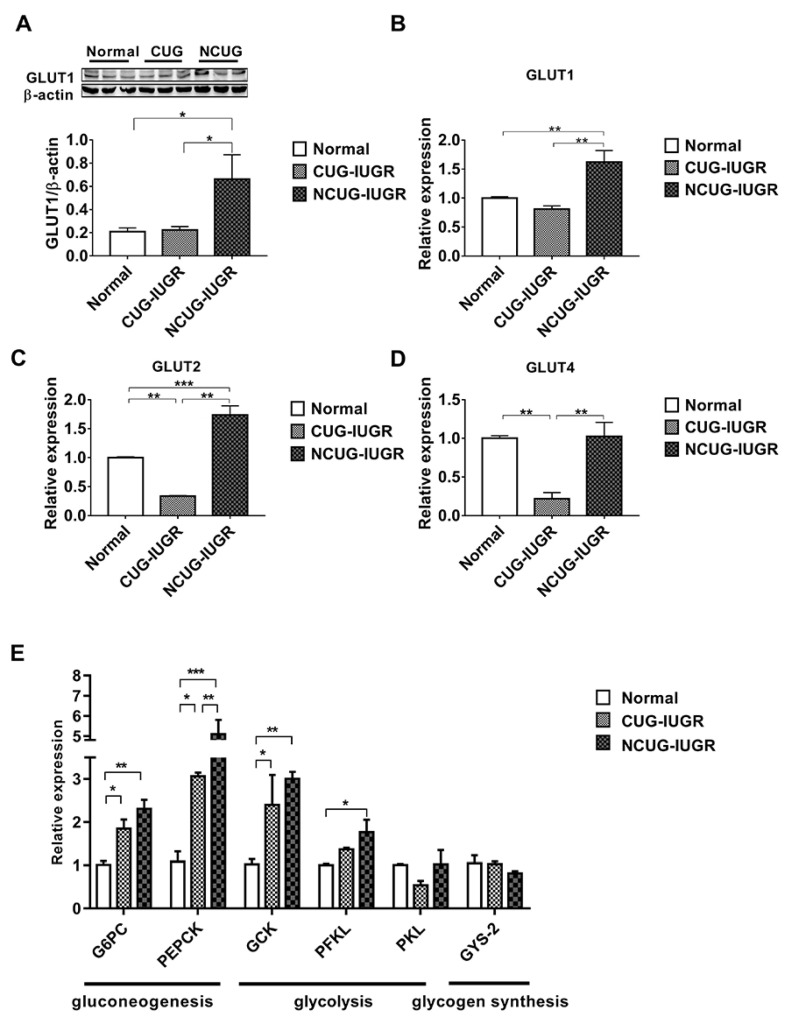
Effects of CUG and NCUG patterns on hepatic glucose metabolism in piglets. (**A**) Protein levels of GLUT1 measured by Western blot analysis. (**B**) mRNA expression of hepatic GLUT1. (**C**) mRNA expression of hepatic GLUT2. (**D**) mRNA expression of hepatic GLUT4. (**E**) Liver glucose metabolism-related mRNA expression. All data with error bars are mean ± SEM. * *p* < 0.05, ** *p* < 0.01, *** *p* < 0.001. CUG = catch-up growth; NCUG = non-catch-up growth; WW = weaning weight.

**Figure 6 metabolites-12-01053-f006:**
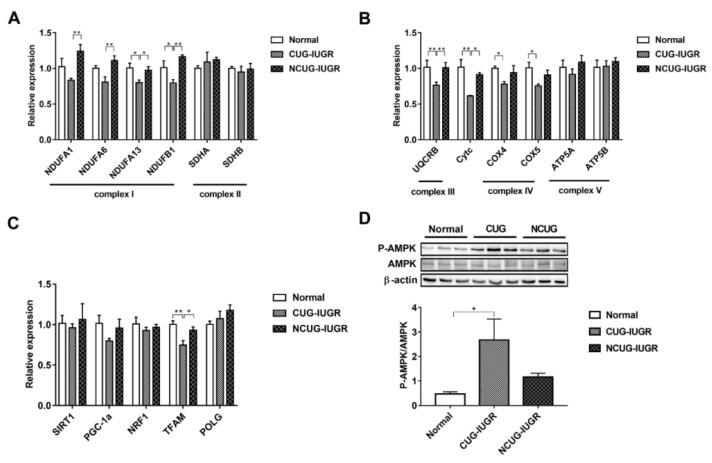
Effects of CUG and NCUG patterns on liver mitochondrial function. (**A**,**B**) mRNA expression of subunits of the liver mitochondrial electron transport chain complex. (**C**) mRNA expression of genes regulates mitochondrial biogenesis. (**D**) Level of AMPK protein phosphorylation. All data with error bars are mean ± SEM. In histograms, * *p* < 0.05, ** *p* < 0.01. CUG = catch-up growth; NCUG = non-catch-up growth; WW = weaning weight.

**Figure 7 metabolites-12-01053-f007:**
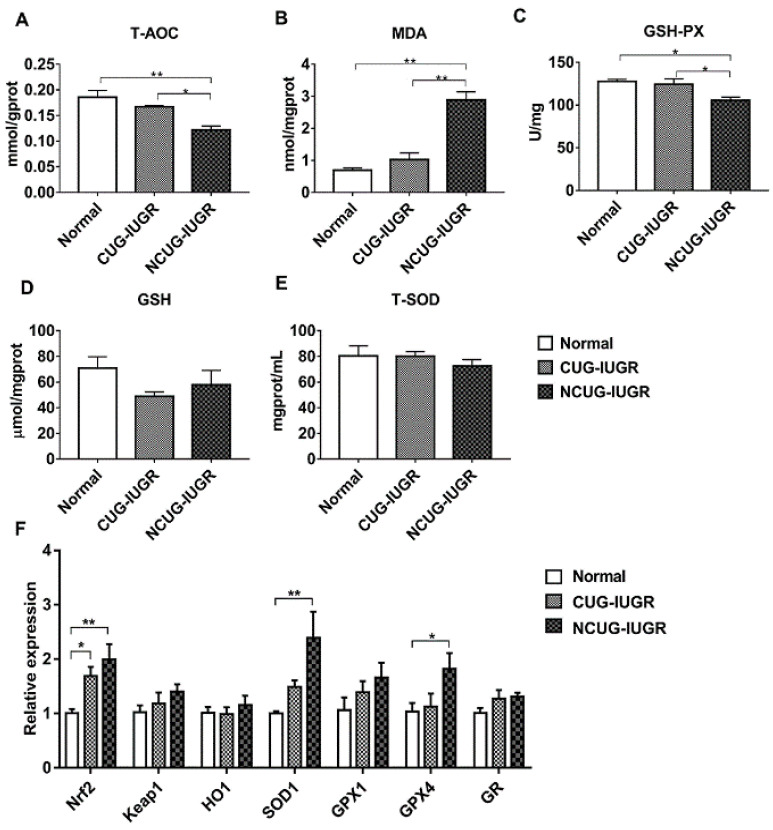
Effects of CUG and NCUG growth patterns on the antioxidant status of the liver. (**A**–**E**) Relative activities of T-AOC, GSH- PX, T-SOD, and relative concentrations of GSH and MDA in liver tissues. (**F**) mRNA expression of liver Nrf2 pathway. All data with error bars are mean ± SEM. In histograms, * *p* < 0.05, ** *p* < 0.0). CUG = catch-up growth; NCUG = non-catch-up growth; WW = weaning weight.

**Figure 8 metabolites-12-01053-f008:**
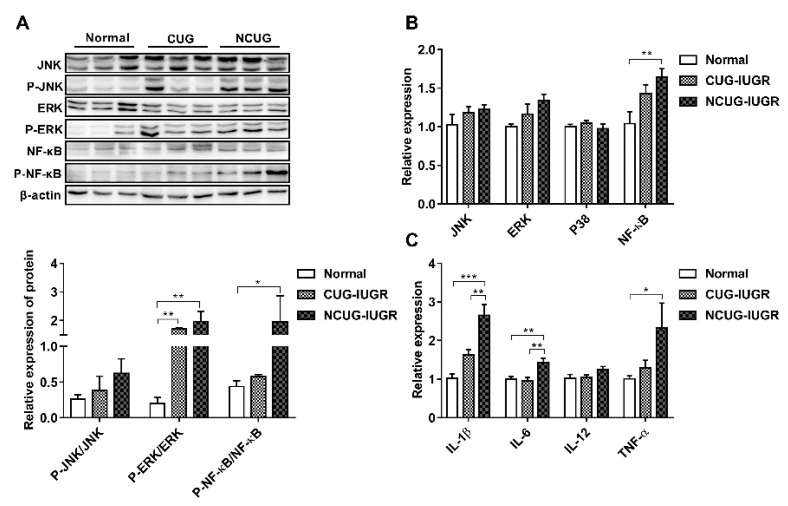
Effects of CUG and NCUG growth patterns on hepatic inflammatory factor expression and *NF-κB*/MAPK pathways. (**A**) Phosphorylation level of the *NF-κB*/MAPK pathway in liver tissue. (**B**) mRNA expression of liver *NF-κB*/MAPK pathway-related genes. (**C**) mRNA expression levels of pro-inflammatory factors in liver tissue. All data with error bars are mean ± SEM. In histograms, * *p* < 0.05, ** *p* < 0.01, *** *p* < 0.001. CUG = catch-up growth; NCUG = non-catch-up growth; WW = weaning weight.

**Figure 9 metabolites-12-01053-f009:**
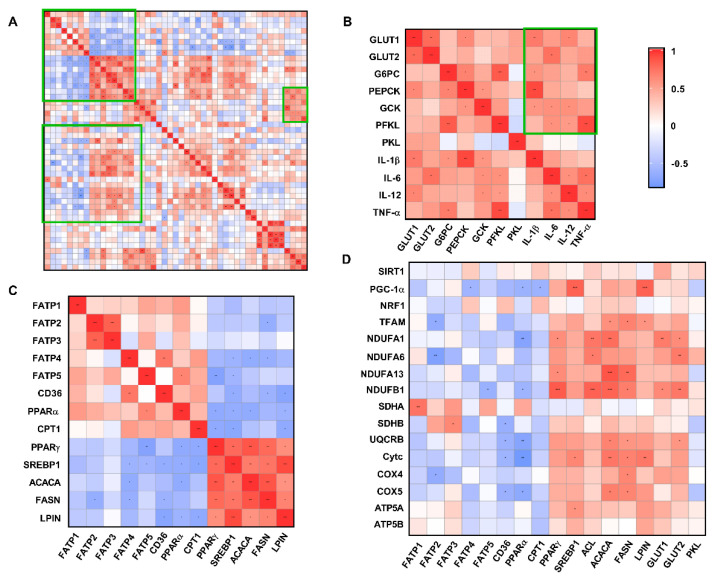
Correlation analysis of liver metabolism-related genes and antioxidant indices (all groups). (**A**) Total correlation analysis of liver metabolism−related genes and antioxidant indices. (**B**) Correlation analysis of mRNA expression levels of glucose metabolism-related genes and pro−inflammatory factors. (**C**) Correlation analysis of mRNA expression levels of genes related to fatty acid experience transport, oxidation, and synthesis. (**D**) Correlation analysis of mRNA expression of glucose and lipid metabolism-related genes and the mRNA expression of mitochondrial function-related genes. * *p* < 0.05, ** *p* < 0.01, *** *p* < 0.001. CUG = catch−up growth; NCUG = non−catch−up growth; WW = weaning weight.

**Table 1 metabolites-12-01053-t001:** Composition and nutrient levels of basal diets (%).

Item	Content	
	Late Gestation	Lactation
Ingredients		
High quality northeast corn	47.90	57.67
Barley	30.00	10.00
Bran	5.00	
Soybean meal (43%)	12.10	22.60
Soybeans		2.00
Fish meal	1.00	1.50
Soybean oil		1.63
Limestone	1.00	1.40
CaHPO_4_	1.20	1.20
Salt	0.50	0.60
Lys	0.15	0.20
DL-Met	0.15	0.20
Premix ^1^	1.00	1.00
Total	100	100
Nutrient composition ^2^		
DE, MJ/kg	3150.00	3400.00
CP	13.50	17.50
EE	2.50	4.80
CF	3.24	2.82
Ash	5.18	5.49
Ca	0.95	1.00
TP	0.60	0.62
AP	0.35	0.45
TLys	0.70	1.20

^1^ The premix provided the following per kg of diets: VA 10,000 IU, VD_3_ 2100 IU, VE 45 IU, VK_3_ 2.0 mg, thiamine 3.0 mg, riboflavin 5.0 mg, VB_6_ 1.8 mg, VB_12_ 0.03 mg, choline chloride 1000 mg, nicotinic acid 25.0 mg, pantothenic acid 15.0 mg, biotin 0.08 mg, folic acid 1.0 mg, Mn 20.0mg, Zn 80.0 mg, Fe 80 mg, Cu 6.0 mg, I 0.15 mg, Se 0.25 mg. ^2^ Nutrient levels are calculated values.

**Table 2 metabolites-12-01053-t002:** Primer sequences used in real-time PCR.

Gene Name	Gene Accession	Primer Sequences (5′ → 3′)	Size, bp
*FATP1*	NM_001083931.1	F: GGCAACAGACGTGATCTATGAC	125
		R: AGCGGCTGGCTGAAAACT	
*FATP2*	JX092264.1	F: TCTAACACGGATGGGGTCG	107
		R: AGGGCAGGAGTGGAAAAGT	
*FATP3*	XM_001929591.2	F: AGGTCTCAGCCGAAGTGGAT	206
		R: TGGGAGGGCGAGGTAGAT	
*FATP4*	XM_003353676.1	F: AGCCGCATCCTGTCCTTT	213
		R: GACATCCTTGGCGATCTTTT	
*FATP5*	AK233227.1	F: GCGAGCAGACGGAAAAGAAG	199
		R: TGCAGGAAGTCCACGAGTGA	
*CD36*	DQ192230.1	F: GGACTCATTGCTGGTGCTGT	169
		R: GTCTGTAAACTTCCGTGCCTGT	
*CPT-1*	NM_001129805	F: GGTGGTGTCAGCGTAGCA	391
		R: CCTTGTTGTCAGTTTGGGTAA	
*PPAR-*α	DQ437887.1	F: CAGCGTGGCACTGAACATC	144
		R: CTCCGATCACATTTGTCATAGAC	
*FASN*	NM_001099930.1	F: GCTTGTCCTGGGAAGAGTGTA	115
		R: AGGAACTCGGACATAGCGG	
*ACACA*	NM_001114269.1	F: ACATCCCCACGCTAAACA	186
		R: AGCCCATCACTTCATCAAAG	
*ACL*	NM_001105302	F: AGCGAGCAGCAGACCTATGAC	143
		R: GGCCACGTTGGTGAAGTTTG	
*LPIN1*	NM_001130734.1	F: CACATTTTGCCCACCCTT	164
		R: GTGCCACGCTCGTTGACC	
*SREBP1*	NM_214157.1	F: AGCGGACGGCTCACAATG	121
		R: CGCAAGACGGCGGATTTA	
*PPAR-*γ	NM_214379.1	F: AGCCCTTTGGTGACTT	213
		R: AGGACTCTGGGTGGTT	
*LXR*	AB254405.1	F: TTCCGTCGCAGTGTCATC	184
		R: CTTGCCGCTTCAGTTTCTTCA	
*GLUT1*	NW_003610563.1	F: GATGAAGGAGGAGTGCCG	106
		R: CAGCACCACGGCGATGAGGAT	
*GLUT2*	NM_001097417.1	F: ATTCTTTGGTGGGATGCTTG	118
		R: ATGAGATGGTCCCAATTTCG	
*GLUT4*	NM_0011288433	F: TATGTTGCGGATGCTATGGG	396
		R: CTCGGGTTTCAGGCACTTTT	
*G6PC*	NM_001113445	F: GGAAATGAGCAGCAAGGT	176
		R: TCGGTGCCACTGATAAAG	
*PEPCK*	NM_001161753	F: CACAAGGGCAAAGTGATTATGC	238
		R: GGAACCAGTTGACGTGGAAGA	
*GCK*	XM_013985832.2	F: GTGGTGGCAATGGTGAATGAC	184
		R: TCGGCGGTCTTCATAGTAGCA	
*PFKL*	XM_021071510.1	F: GAAACGAGAAGTGCCACGAA	182
		R: TACCGTAGTTCCGGTCGAAG	
*PKL*	XM_021089721.1	F: AGACTGCCAAGGGTCACTTT	117
		R: CAGCTCCTCAAAGAGTTGCC	
*GYS2*	NM_001195511	F: CATCACCACCAACGACGGA	193
		R: ACACGGCCCAGAGAAAAGG	
*NDUFA1*	XM_003135339.5	F: GCTTCCGGGGAAGGAATCAA	101
		R: CCGGGGAGAATTTCGAACCA	
*NDUFA6*	NM_001185178.1	F: TCTCAGAGCCTTGCATGTCG	85
		R: AAGCCATCCAGCATCGTACC	
*NDUFA13*	NM_001244646.1	F: ATGAAGGATGTGCCGGACTG	125
		R: CCATAGGTGGCGCTGAGAAT	
*NDUFB1*	XM_003482306.4	F: TGCCTTCCGGAACAAGAGTC	88
		R: GCAATTCAGCCACAGCCTTT	
*SDHA*	XM_021076930.1	F: CAATAAGAGGTCGTCGGCCA	127
		R: AGAGAGACCAAACGCAGCTC	
*SDHB*	NM_001104953.1	F: TCCTATGGTGTTGGATGCGT	124
		R: AGTGTTGCCTCCGTTGATGT	
*UQCRB*	NM001185172.1	F: CATCAGGCAACGCTTCTGTC	81
		R: TATACCCTCCAGCCACTTGC	
*CytC*	NM_001129970.1	F: CTGGGGAGAGGAGACACTGAT	158
		R: AGGCGGTGGCCAACTTTTAC	
*COX IV*	XM_021093705.1	F: CCAAGTGGGACTACGACAAGAAC	131
		R: CCTGCTCGTTTATTAGCACTGG	
*COX V*	NM_001007517.1	F: ATCTGGAGGTGGTGTTCCTACTG	160
		R: GTTGGTGATGGAGGGGACTAAA	
*ATP5A1*	NM_001185142.1	F: ACGCCATTGATGGAAAGGGT	98
		R: TGGTTCCCGCACAGAGATTC	
*ATP5B1*	XM_001929410.5	F: CATGTTGGGCTTTGTGGGTC	139
		R: ATAGTCTCTGGCAGGCTGGA	
*SIRT1*	NM_001145750.2	F: TTGCAACAGCATCTTGCCTG	91
		R: GGACATCGAGGAACCACCTG	
*PGC-1α*	NM_213963.2	F: GCTTGACGAGCGTCATTCAG	100
		R: GGTCTTCACCAACCAGAGCA	
*NRF1*	XM_021078993.1	F: GAAGCTGTCCAGGGGCTTTA	116
		R: ATCCATGCTCTGCTACTGGG	
*TFAM*	NM_001130211.1	F: AGCGAGGTCTGAAGAGTTGC	114
		R: TTGCACCCGTAGACAAAGCA	
*POLG*	XM_001927064.5	F: CTGTCAGATGAGGGCGAGTG	133
		R: ACTTCTTCCGTCGTGACTTTCT	
*Nrf2*	XM_013984303.2	F: ATCCAGCGGATTGCTCGTAG	155
		R: TCAAATCCATGTCCTTGGCG	
*Keap1*	NM_001114671.1	F: TCTGCTTAGTCATGGTGACCT	158
		R: GGGGTTCCAGATGACAAGGG	
*HO-1*	NM_001004027.1	F: TGATGGCGTCCTTGTACCAC	71
		R: GACCGGGTTCTCCTTGTTGT	
*SOD1*	NM_214127.2	F: CAAGAAGGGGCACCACGTT	70
		R: CTCAGGGGACGCAAGAACTG	
*GPX1*	NM_214201.1	F: CCTCAAGTACGTCCGACCAG	85
		R: GTGAGCATTTGCGCCATTCA	
*GPX4*	NM_214407.1	F: TGTGTGAATGGGGACGATGC	135
		R: CTTCACCACACAGCCGTTCT	
*GR*	AY368271.1	F: GTGAGCCGACTGAACACCAT	141
		R: CAGGATGTGAGGAGCTGTGT	
*TNF-*α	NM_214022.1	F: GCCCTTCCACCAACGTTTTC	97
		R: CAAGGGCTCTTGATGGCAGA	
*IL-1β*	NM_214055.1	F: ATTCAGGGACCCTACCCTCTC	92
		R: ATCACTTCCTTGGCGGGTTC	
*IL-6*	NM_214399.1	F: ACAAAGCCACCACCCCTAAC	185
		R: CGTGGACGGCATCAATCTCA	
*IL-12*	NC_010458.4	F: CAACCCTGTGCCTTAGCAGT	113
		R: AGAGCCTGCATCAGCTCAGT	
*NF-κB*	NM_001114281.1	F: GGGGCGATGAGATCTTCCTG	110
		R: CACGTCGGCTTGTGAAAAGG	
*ERK*	XM_021088019.1	F: CAGTCTCTGCCCTCCAAGAC	139
		R: GGGTAGATCATCCAGCTCCA	
*JNK*	XM_001929166.6	F: TGGATGAAAGGGAACACACA	104
		R: ATGATGACGATGGATGCTGA	
*P38*	XM_021091323.1	F: AAGACGGGGTCCTCATCTCC	230
		R: TCTCATCGTAGGGCTCTGCT	

**Table 3 metabolites-12-01053-t003:** Growth performance of piglet.

Item	Normal	CUG	NCUG	SEM	*p*-Value
BW, kg	1.388 *^a^* ± 0.010	1.059 *^b^* ± 0.035	1.045 *^b^* ± 0.037	0.0203	<0.001
WW, kg	7.530 *^a^* ± 0.526	7.457 *^a^* ± 0.368	5.592 *^b^* ± 0.912	0.1367	<0.001
ADG, kg	0.236 *^a^* ± 0.017	0.246 *^a^* ± 0.014	0.175 *^b^* ± 0.035	0.0048	<0.001

Note: CUG = catch-up growth; NCUG = non-catch-up growth; WW = weaning weight. All data shown as means ± SD, values with different small letter superscripts differ (*p* < 0.001).

**Table 4 metabolites-12-01053-t004:** Weekly average daily gain of piglets.

ADG Per Week, kg	Normal	CUG	NCUG	SEM	*p*-Value
1 d–7 d	0.168 *^a^* ± 0.019	0.163 *^a^* ± 0.026	0.126 *^b^* ± 0.042	0.004	<0.001
7 d–14 d	0.204 *^a^* ± 0.064	0.241 *^a^* ± 0.046	0.161 *^b^* ± 0.069	0.007	<0.001
14 d–21 d	0.228 *^b^* ± 0.017	0.264 *^a^* ± 0.044	0.198 *^c^* ± 0.045	0.005	<0.001
21 d–26 d	0.343 *^a^* ± 0.031	0.346 *^a^* ± 0.035	0.231 *^b^* ± 0.077	0.009	<0.001

Note: CUG = catch-up growth; NCUG = non-catch-up growth; WW = weaning weight. All data are shown as means ± SD, values with different small letter superscripts differ (*p* < 0.001).

**Table 5 metabolites-12-01053-t005:** Organ weight of piglets.

Organ Weight, kg	Normal	CUG	NCUG	SEM	*p*-Value
Heart	0.093 *^a^* ± 0.011	0.097 *^a^* ± 0.006	0.057 *^b^* ± 0.006	0.007	0.02
Liver	0.447 *^a^* ± 0.025	0.397 *^b^* ± 0.030	0.280 *^c^* ± 0.017	0.020	<0.001
Kidney	0.090 *^a^* ± 0.010	0.087 *^a^* ± 0.006	0.067 *^b^* ± 0.006	0.006	0.017
Spleen	0.047 *^a^* ± 0.047	0.033 *^ab^* ± 0.033	0.020 *^b^* ± 0.020	0.008	0.037
Lung	0.220 *^a^* ± 0.046	0.180 *^a^* ± 0.010	0.120 *^b^*	0.022	0.011
Pancreas	0.010	0.013 ± 0.006	0.013 ± 0.006	0.004	0.63

Note: CUG = catch-up growth; NCUG = non-catch-up growth; WW = weaning weight. All data are shown as means ± SD, and values with different small letter superscripts differ (*p* < 0.05).

**Table 6 metabolites-12-01053-t006:** Organ index of piglets.

Organ Index, %	Normal	CUG	NCUG	SEM	*p*-Value
Heart	1.15 ± 0.17	1.19 ± 0.050	1.18 ± 0.150	0.110	0.932
Liver	5.49 *^ab^* ± 0.436	4.86 *^b^* ± 0.303	5.83 *^a^* ± 0.212	0.270	0.031
Kidney	1.10 *^a^* ± 0.096	1.07 *^b^* ± 0.084	1.39 *^a^* ± 0.159	0.096	0.028
Spleen	0.58 ± 0.200	0.41 ± 0.066	0.42 ± 0.013	0.010	0.245
Lung	2.71 ± 0.610	2.21 ± 0.167	2.50 ± 0.081	0.300	0.322
Pancreas	0.12 ± 0.003	0.16 ± 0.069	0.28 ± 0.120	0.065	0.123

Note: CUG = catch-up growth; NCUG = non-catch-up growth; WW = weaning weight. All data are shown as means ± SD, and values with different small letter superscripts differ (*p* < 0.05).

## Data Availability

The data presented in this study are available in the article.

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
