# Peer review of "Altered Liver Metabolism, Mitochondrial Function, Oxidative Status, and Inflammatory Response in Intrauterine Growth Restriction Piglets with Different Growth Patterns before Weaning"

_metabolites, 2022, doi:10.3390/metabo12111053_

Round 1
Reviewer 1 Report
The paper by Dr. Jun Wang describes intrauterine growth restriction (IUGR) caused liver damage through alteration of metabolism. This study showed lipid accumulation was developed however mitochondrial function, oxidative stress and inflammatory response were partially restored in liver of catch-up growth (CUG). It contributes intervention of liver dysfunction in IUGR. The authors have some novel data but some of the methods need to be clarified.
1. Please add exact reference. (line 72-75)
2. It is unclear the number exactly show in Figure 2D. X axis might represent survival rate not mortality rate. Why the percentage was lower at 26d than at 21d if the number show survival rate. Please correct it.
3. Please add description of figure 8a in text.
4. It is unclear how calculate and where the data come from in figure 9. Is that RNA-seq data? Which group was used for it?
5. GUG might be CUG. Please correct it. (line 401)
6. Please clarify why lipid accumulation was developed in CUG although AMPK activation and PPARα gene expression were enhanced.
Reviewer 2 Report
All abbreviation should be explained at the first quotation. The abbreviation in the title should also be explained.
Please add the number and date of receipt of the consent from ethical commission.
Line 110 The phrase “animals were euthanized” would be more appropriate.
In specifications of reagents and equipment, the name of firm, city and country should be given. The authors often omit cities
Scale bars on the microphotographs are completely unreadable. Please re-edit them.
A list of abbreviations added at the beginning of the article would significantly increase its readability.
Whether the study in opinion of authors has any limitations. If yes, please add them in the discussion
whether the research is of practical importance for pig breeding. If so, it is worth mentioning it in the conclusions.
Reviewer 3 Report
The study by Wang et al. is very well design and written. The problem of intrauterine growth restriction (IUGR) in pigs is very important problem, thus the undertaken research is of high intrest and importance. The methods used are adequte and sufficient to verify the assumed hypothesis of the study. My only comments are related to the number of approval of ethic committee - it should be provide. Furthermore, the catalog numbers of kits for hepatic oxidative stress mesurement should be added.
Round 2
Reviewer 1 Report
The authors have made substantial revisions in response to the original critique. However the authors need resolve a problem.
1. What is the index parameters? Did they come from RT-PCR? Did they consist of all group data? Please clarify it in method section and figure legend.
